# No Racial Disparities Observed Using Point-of-Care Genetic Counseling and Testing for Endometrial and Ovarian Cancer in a Diverse Patient Population: A Retrospective Cohort Study

**DOI:** 10.3390/cancers16081598

**Published:** 2024-04-22

**Authors:** Michael Kim, Judy Hayek, Cheyenne Acker, Anjile An, Peilin Zhang, Constantine Gorelick, Margaux J. Kanis

**Affiliations:** 1New York-Presbyterian Brooklyn Methodist Hospital, Brooklyn, NY 11215, USA; 2Division of Gynecologic Oncology, State University of New York Downstate Medical Center, Brooklyn, NY 11203, USA; 3Weill Cornell Medicine, New York, NY 10065, USA

**Keywords:** genetic testing, genetic counseling, hereditary cancer, ovarian cancer, endometrial cancer

## Abstract

**Simple Summary:**

Despite consensus recommendations, national genetic counseling and testing rates for gynecological cancers remain relatively low. Prior studies have also shown that referrals are fewer for people of minority backgrounds. This initial study aimed to determine what genetic counseling and testing rates are found in a diverse patient setting. High rates of genetic counseling and testing are demonstrated, with no racial disparities. Both endometrial and ovarian cancer data are presented simultaneously in a single study in the first of its kind, helping provide a broader combined perspective. Point-of-care genetic counseling and testing by gynecologic oncologists is a strategy that can be similarly adopted at other institutions to potentially reduce disparities. Further studies can identify other factors responsible for the successful counseling and testing rates and how they can be used to broaden the reach of these services at other centers with similar diverse populations.

**Abstract:**

We investigated genetic counseling and testing rates for patients with gynecologic malignancy at a tertiary care center with a large minority population. Our retrospective cohort included newly diagnosed epithelial ovarian, fallopian tube, peritoneal, or endometrial cancer patients between January 2014 and June 2022. For endometrial cancer, 373 patients were identified. A total of 207 (55%) patients were screened using mismatch repair immunohistochemistry (MMR IHC). A total of 82 (40%) had MMR deficiencies on IHC. Of these, 63 (77%) received genetic counseling. A total of 62 (98%) underwent genetic testing, and ultimately, 7 (11%) were diagnosed with Lynch syndrome (LS). The overall rate of LS was 1.9%. MMR IHC testing increased steadily, reaching 100% in 2022. For ovarian cancer, 144 patients were identified. A total of 104 (72%) patients received genetic counseling, and 99 (95%) underwent genetic testing. Rates were not influenced by race, ethnicity, insurance type, or family history of cancer. They were significantly different by cancer stage (*p* < 0.01). The proportion of patients who received genetic counseling increased from 47% in 2015 to 100% in 2022 (*p* < 0.01). Most counseling was performed by a gynecologic oncologist (93%) as opposed to a genetic counselor (6.7%). Overall, 12 (8.3%) patients were *BRCA+*. High rates of counseling and testing were observed with few disparities.

## 1. Introduction

In 2014, the Society of Gynecologic Oncology (SGO) released new guidance expanding the role of genetic testing within gynecologic cancers. Women diagnosed with endometrial cancer should be evaluated for Lynch syndrome by molecular screening when feasible or through a systematic review of personal and familial history [1]. For example, women with a family history of colon cancer or endometrial cancer should undergo genetic counseling, irrespective of personal history. Women diagnosed with epithelial ovarian, tubal, and peritoneal cancers, regardless of family history or age, should receive genetic counseling with the option of genetic testing [1].

Endometrial cancer is the most common gynecologic cancer in the United States, with an estimated 66,200 cases diagnosed in the country in 2023 [2]. Up to 5% of endometrial cancer can be attributed to hereditary causes, the majority of which are related to Lynch syndrome (LS) [3].

LS is an autosomal dominant disorder characterized by germline variants in DNA mismatch repair (MMR) genes, including *MLH1*, *MSH2*, *MSH6*, *PMS2*, and *EPCAM*. LS raises susceptibility to a variety of cancers [3,4]. Most notably, it raises colorectal cancer lifetime risk to 40–80% and endometrial cancer to 33–61% [5]. Endometrial cancer is the sentinel cancer for approximately 50% of women with LS [3,6]. The identification of LS in these patients is critical in effective cancer screening, risk-reducing strategies, and cascade testing [3,5,6]. For example, an LS diagnosis with appropriate colonoscopy surveillance leads to a 60% reduction in colorectal cancer incidence and up to a 70% reduction in colorectal cancer mortality [5]. Knowledge of LS also allows for the use of targeted therapies such as pembrolizumab [3,4].

Ovarian cancer is the second most common gynecologic cancer in the United States, with an estimated 19,710 cases to be diagnosed in the United States in 2023 [2]. The *BRCA* variant is the strongest risk factor for the development of ovarian cancer, accounting for up to 15–20% of cases [4]. While the lifetime risk of ovarian cancer in the general population is 1.3% [4,7], women with *BRCA1* variants have a lifetime risk of as high as 39–49% and up to 11–18% for *BRCA2* [7].

Patients with *BRCA1/2* variants have a higher risk than the general population of a variety of cancers, most notably breast cancer (40–80%) [8]. Additionally, genetic testing can reveal homologous recombination deficiencies (HRD), which can be as high as 30% in epithelial ovarian cancer [9]. Knowledge of *BRCA* and HRD status allows for the use of targeted therapies for ovarian cancer, such as poly(ADP)ribose polymerase inhibitors [4,7,8,10,11].

With improved genetic sequencing, expanding insurance coverage, and wider institutional adaptation, genetic testing has become more accessible [12]. Despite the professional recommendations and known advantages in the realm of gynecologic cancers, rates remain low [3,7,8,10,13]. National trends for genetic counseling referral for ovarian cancer patients range from 12 to 30% [7]. At large academic institutions with in-house genetic experts, genetic testing rates in ovarian cancer are 53–74% [7,11].

Significant disparities exist within gynecologic genetic testing. Significantly higher rates have been associated with White race [7], private insurance [7,10], English as a primary language [7], younger age of diagnosis [3], a lower BMI [3], higher-grade serous histology [7], a family history of cancer [3,10,11], and a personal history of cancer [3,10]. Lower rates have been associated with Black race [6], Hispanic ethnicity [6], older age of diagnosis [11], a primary language other than English [7], and public insurance [7,10].

This study aims to characterize genetic counseling and testing patterns in patients diagnosed with ovarian or endometrial cancer at a tertiary care center with a large minority population.

## 2. Materials and Methods

This is a two-part retrospective cohort study of patients newly diagnosed with either endometrial cancer or epithelial ovarian, fallopian tube, or peritoneal cancer between January 2014 and June 2022, as confirmed by surgical pathology. Data were obtained from pathology and chemotherapy databases at the NewYork-Presbyterian Brooklyn Methodist Hospital. The study protocol was approved by the institutional review board. Study data were collected and managed using REDCap electronic data capture tool (REDCap 14.0.14) [14,15].

A total of 315 patients with endometrial adenocarcinomas were identified from the pathology department database, and 119 were identified from the infusion center database for a total of 434 non-overlapping patients. Of these, 61 patients were excluded due to incomplete data, with reasons including early transfer of care, loss to follow-up, missing encounter notes, or pathology with concurrent sarcomas or other rare malignancies. Similarly, 75 patients with epithelial ovarian, fallopian tube, and primary peritoneal cancer were extracted from the pathology department database, and 124 similarly from the infusion center for a total of 199 patients identified. A total of 55 patients were excluded due to reasons same as noted above for endometrial cancer.

Variables including age (years), race (Asian, Black, Other, White), ethnicity (Hispanic, non-Hispanic), BMI (kg/m^2^), insurance (private, Medicare, Medicaid, other/unknown), language (English, non-English), family history of cancer, date of diagnosis (year), stage (I, II, III, IV), MMR immunohistochemistry status, *BRCA1/2* and LS status, and genetic counseling/testing status were manually abstracted. “Ethnicity”, often used variably in the literature, as collected within our electronic health record system, was binary, either Hispanic or non-Hispanic [16,17]. Point-of-care genetic counseling and testing were initially implemented for those eligible, either by a gynecologic oncologist or a trained genetic counselor. Patients who screened positive on subsequent genetic testing by a medical provider were further referred for additional counseling by a genetic counselor. Genetic testing performed at this institution involved external third party vendors.

Descriptive statistics were used for the study sample with respect to clinical variables of interest. The chi-square test or Fisher’s exact test was used to examine the association between genetic counseling/testing and categorical variables. Two-sample *t*-test or Wilcoxon rank-sum test was used for the association between continuous factors. The Mann–Kendall trend test was used to determine the significance of year-to-year trends.

In accordance with the journal’s guidelines, we will provide our data for independent analysis by a selected team by the Editorial Team for the purposes of additional data analysis or for the reproducibility of this study in other centers if such is requested.

## 3. Results

### 3.1. Endometrial Cancer

In our cohort, 373 patients with endometrial cancer were encountered. A total of 45% identified as White, 42% as Black, 1.3% as Asian, and 12% as other/unknown; 8.3% were of Hispanic ethnicity, and 18% were non-English speaking. The mean age at diagnosis was 66 years (SD10). A total of 207 (55%) patients were screened using MMR IHC. A total of 82 (40%) of these patients had MMR deficiencies on IHC. Of these, 63 (77%) received genetic counseling. Sixty-two (98%) of those counseled subsequently underwent genetic testing, and ultimately, 7 (11%) were diagnosed with LS. The overall rate of LS detected in the study population was 1.9% (Table 1, Figure 1).

MMR IHC screening rates were influenced by the mean age at diagnosis (*p* < 0.01) and insurance type (*p* = 0.04). The mean age of MMR completed was 64 (SD10), and that of MMR not completed was 67 (SD10). A total of 45% of patients who had MMR IHC applied had private insurance, compared to 34% of patients who did not. This rate was not influenced by race, language, BMI, family history of cancer, or stage. Over the study period, the rate of MMR IHC testing was noted to increase, approaching 95% in 2021 and 100% in 2022 (*p* = 0.00058) (Figure 2).

The proportion of patients who received genetic counseling and testing for endometrial cancer also showed a significant upward trend over time (*p* < 0.01) (Figure 3). Genetic counseling and testing rates significantly varied with different ethnicities (*p* = 0.03), with only 3.0% of patients receiving genetic counseling/testing identifying as Hispanic. A total of 98% of initial genetic counseling was performed by a gynecologic oncologist, as opposed to a genetic counselor (*p* < 0.001).

### 3.2. Ovarian Cancer

A total of 144 patients with epithelial ovarian, fallopian tube, or peritoneal cancer were identified. The mean age at diagnosis was 63 years (SD13). This diverse cohort included 51% White, 36% Black, 3.5% Asian, and 9% other/unknown; 9% were of Hispanic ethnicity, and 26% were non-English speaking (Table 1).

Of the 144 patients, 104 (72%) patients received genetic counseling, and 99 (69%) received genetic testing (Figure 4). Hence, 95% of those who underwent genetic counseling also underwent testing. The genetic counseling and testing rates were not influenced by race, ethnicity, language, insurance type, BMI, or a family history of cancer. Genetic testing was associated with significant differences by cancer stage (*p* = 0.001)

There was a significant upward trend toward a higher proportion of patients receiving genetic counseling, from 47% in 2015 to100% in 2022 (*p* < 0.01) (Figure 5). Most of the initial genetic counseling was performed by a gynecologic oncologist (93%) as opposed to a genetic counselor (6.7%). Overall, 12 (8.3%) patients were *BRCA*+. Up to 90% of patients did not pay any additional out-of-pocket expenses to receive testing.

## 4. Discussion

Per the Society of Gynecologic Oncology recommendations, women diagnosed with endometrial cancer should be screened for Lynch syndrome by molecular screening when feasible, and women diagnosed with epithelial ovarian, tubal, and peritoneal cancers, regardless of family history or age, should receive genetic counseling with the option of genetic testing [1].

### 4.1. Summary of Main Results

For those diagnosed with endometrial cancer, we found that 207 (55%) patients were screened using MMR IHC staining of the pathology specimens. The rate increased annually, reaching 95% in 2021 and 100% in 2022, likely attributable to wider institutional availability of MMR IHC beginning in 2016. We also showed that 77% of those with MMR deficiencies received genetic counseling.

Furthermore, we observed that IHC screening rates may be affected by insurance type: 45% of patients who received screening had private insurance, compared to 34% of patients who did not. There were also differences in genetic counseling and testing rates based on ethnicity, with only 3% (n = 3) of those who received these services identifying as Hispanic. It should be noted that when stratified by race, however, there were no differences. The rate of LS detected in our cohort was 1.9%, which is less than the known prevalence in endometrial cancer of 2–5% [18,19].

On the other hand, the genetic counseling rate for ovarian, fallopian tube, or primary peritoneal cancer (OC) at our institution was 72%, with the percentage of people receiving counseling increasing yearly, from 47% in 2015 to 100% in 2022. Additionally, genetic counseling and testing rates for OC in our study were not influenced by race, ethnicity, language, insurance type, BMI, or a family history of cancer. The overall *BRCA*-positive rate was 8.3%.

### 4.2. Results in the Context of Published Literature

For endometrial cancer, in comparison to the 207 (55%) patients screened for MMR IHC at our institution, reaching 99% in 2021 and 100% in 2022, a previous cohort study by Huang et al. at the University of Miami similarly showed that 51.1% of endometrial cancer patients received IHC screening between 2014 and 2017 [18].

Furthermore, while 77% of those with MMR deficiencies received genetic counseling at our institution, a prior study by Lee et al. at New York University showed only 58% of women with high-risk characteristics for LS, such as age < 50 at the time of diagnosis, two or more family members with LS cancers, a metachronous or synchronous LS cancer, or evidence of MMR loss on IHC, were referred for genetic counseling [3]. 

Additionally, while we found that Hispanic patients and those without private insurance may be less likely to receive genetic counseling and testing, Huang et al. demonstrated higher rates of screening for Hispanic women and those reliant on Medicaid compared to those with Medicare or private insurance [18]. Overall, studies addressing disparities in endometrial cancer genetic screening remain relatively rare when compared with ovarian cancer. 

In a 2019 systematic review including 12,633 patients with endometrial cancer, it was shown that approximately 3% of endometrial cancers can be attributed to LS, similar to rates in colorectal cancer patients [19]. Our lower rate of 1.9% may be attributed to possible demographic differences or gaps in screening. These findings are thought-provoking since, at our institution, universal IHC screening is implemented. Interestingly, a retrospective study of endometrial cancer cases at Kaiser Permanente Northern and Southern California involving 2045 patients showed that risk-based, physician-ordered IHC screening, when compared with universal, automated IHC screening, showed no difference in LS cases detected [20].

Of those diagnosed with OC at our institution, 72% received subsequent genetic counseling, increasing to 100% in 2022. By comparison, national trends for genetic counseling referral for epithelial ovarian cancer patients range from 12–30% [7]. At large academic institutions with in-house genetic experts, genetic testing rates in ovarian cancer were higher, at 53–74% [7,11]. In a meta-analysis by Lin et al. in 2021 involving 35 studies and 16,891 patients with ovarian cancer, the rate of referral to genetic counseling and completion of genetic testing were 39% and 30%, respectively [13].

Furthermore, while our genetic counseling and testing rates for OC were not influenced by demographic variables, a prior study by Manrriquez et al. at the University of California, San Francisco, discovered that English as a primary language and private insurance/Medicare were predictive of more genetic counseling referrals, with lower rates reported for minority women and those with public insurance. Only one-third of black women in their study were referred for genetic counseling, and none received counseling [7]. Lin et al., in their meta-analysis, also reported that Black race was associated with lower rates of testing compared to White (26% vs. 40%) and uninsured versus insured (23% vs. 38%) [13]. 

Lastly, our *BRCA*-positive rate of 8.3% is lower than the known rate of 15–20% in epithelial ovarian cancers [4], also indicating possible gaps in screening or demographic differences. 

### 4.3. Strengths and Weaknesses

One limitation of our findings is that, as a cohort study, we can only describe associations and not causation. As a single institution study, we are also limited in how generalizable our results will be to the wider population. A notable strength of our study is the combination of both endometrial and ovarian cancer genetic counseling and testing rates at an institution into a single manuscript. To our knowledge, this is the first of its kind to do so. Our study contributes to the discussion on broadening access to genetic counseling and testing for gynecologic cancers, particularly in similar racially/ethnically diverse patient populations. 

### 4.4. Implications for Practice and Future Research

Prior studies have demonstrated that oncologist-led genetic counseling and testing is a possible model to increase access to these services, particularly in low-resource settings [21,22,23]. However, given the complexity of genetic testing and the possible implications of both negative and positive results, it is important to recognize the responsibility associated with genetic counseling and the expertise it requires. “Mainstreaming” genetic counseling to non-genetic counseling specialists requires that the surgeon or oncologist becomes versed in the subject matter, and additional training may be necessary [23,24,25,26]. Point-of-care counseling and testing were likely among the most important factors that contributed to the high rates at our institution.

Future studies can further identify other factors that are responsible for the high rates of counseling and testing at our institution despite having a diverse patient demographic with relatively low socioeconomic status. Studies can also investigate any barriers that exist, if any, particularly for Hispanic patients with endometrial cancer [27,28,29].

## 5. Conclusions

We confirmed that relatively high levels of counseling and testing can be performed in a diverse patient population using point-of-care counseling and testing. We also observed much fewer disparities in access to these services, in stark contrast to previous studies. When it came to ovarian cancer, for example, there were no disparities at our institution based on race, ethnicity, language, or insurance type. This initial study provides the necessary foundation for future investigation into strategies that can broaden access to these services.

## Figures and Tables

**Figure 1 cancers-16-01598-f001:**
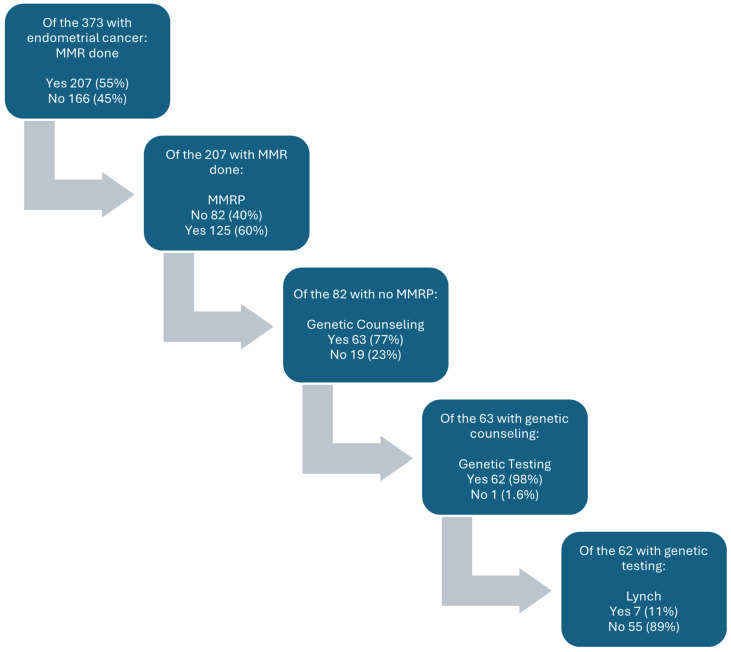
Breakdown of endometrial cancer counseling and testing.

**Figure 2 cancers-16-01598-f002:**
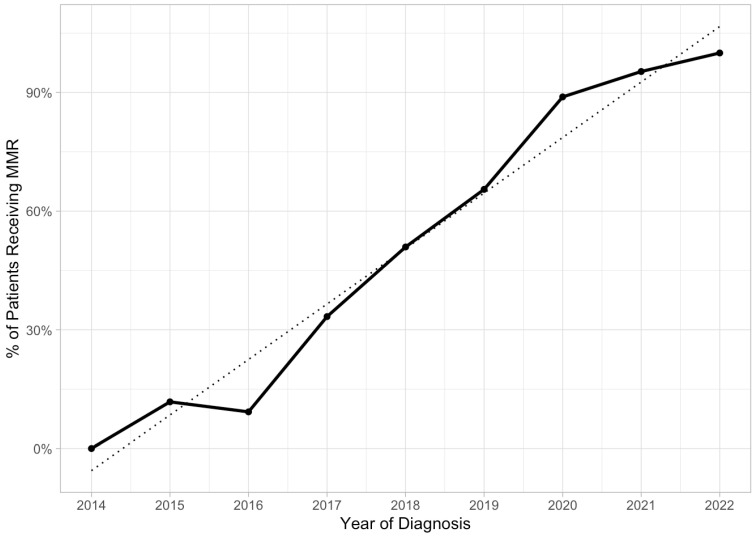
Trend over time of proportion of patients who received MMR testing following diagnosis of endometrial cancer. The Mann–Kendall trend test for proportion of patients receiving MMR indicated a statistically significant positive/upward trend (tau = 0.944, *p*-value = 0.00058).

**Figure 3 cancers-16-01598-f003:**
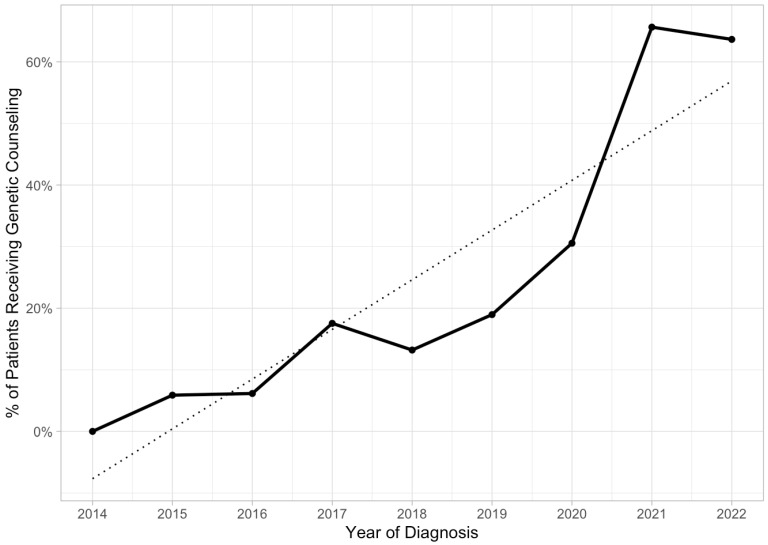
Trend over time of proportion of patients who received genetic counseling following diagnosis of endometrial cancer. The Mann–Kendall trend test for proportion of patients receiving MMR indicated a statistically significant positive/upward trend (tau = 0.889, *p*-value = 0.00123).

**Figure 4 cancers-16-01598-f004:**
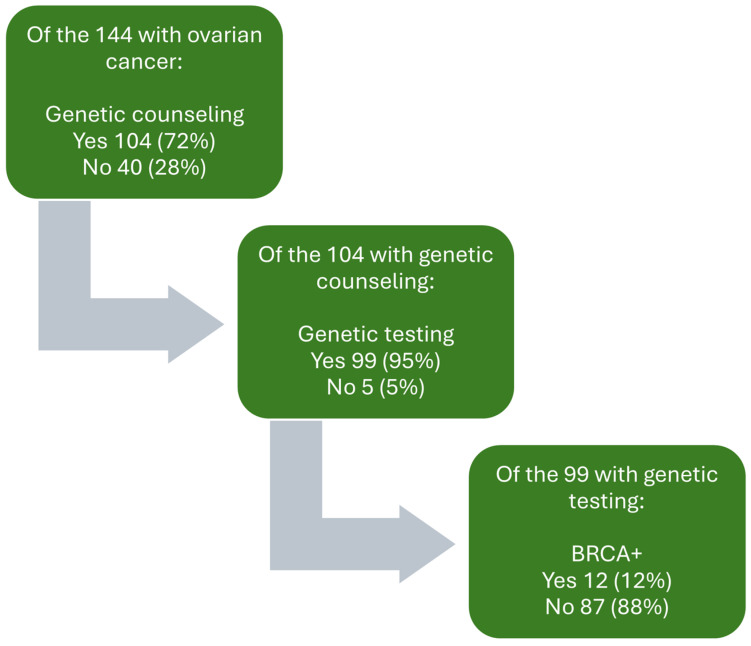
Breakdown of ovarian cancer counseling and testing.

**Figure 5 cancers-16-01598-f005:**
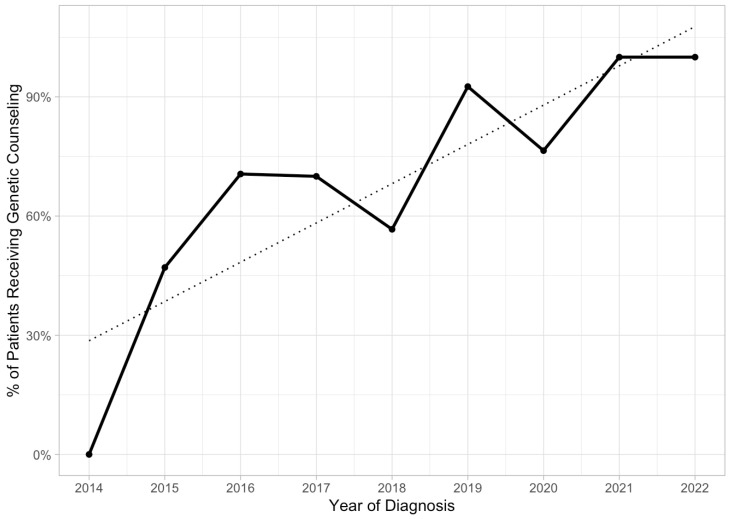
Trend over time of proportion of patients who received genetic counseling after being diagnosed with epithelial ovarian, fallopian tube, or primary peritoneal cancer. The Mann–Kendall trend test for the proportion of patients receiving genetic counseling indicated a statistically significant positive/upward trend (tau = 0.761, *p*-value = 0.00642).

**Table 1 cancers-16-01598-t001:** Descriptive statistics of those with endometrial and ovarian cancer, respectively.

Characteristic	Endometrial	Ovarian
Year of Diagnosis	N = 373	N = 144
2014	1 (0.3%)	1 (0.7%)
2015	17 (4.6%)	17 (12%)
2016	65 (17%)	17 (12%)
2017	57 (15%)	20 (14%)
2018	53 (14%)	30 (21%)
2019	58 (16%)	27 (19%)
2020	36 (9.7%)	17 (12%)
2021	64 (17%)	7 (4.9%)
2022	22 (5.9%)	8 (5.6%)
Age at Diagnosis		
Mean (SD)	66 (10)	63 (13)
Range	31, 92	19, 93
Race		
Asian	5 (1.3%)	5 (3.5%)
Black	155 (42%)	52 (36%)
Other/Unknown	46 (12%)	13 (9.0%)
White	167 (45%)	74 (51%)
Ethnicity		
Hispanic	31 (8.3%)	13 (9.0%)
Non-Hispanic	342 (92%)	131 (91%)
Language		
English	306 (82%)	107 (74%)
Non-English	67 (18%)	37 (26%)
Insurance		
Private	149 (40%)	65 (45%)
Medicare	198 (53%)	62 (43%)
Medicaid/None	24 (6.4%)	17 (12%)
Other/Unknown	2 (0.5%)	0 (0%)
BMI		
Mean (SD)	34 (9)	28.2 (6.8)
Range	16, 61	16.0, 66.1
Family History of Cancer	68 (18%)	72 (50%)
Stage		
I	233 (62%)	25 (17%)
II	32 (8.6%)	11 (7.6%)
III	61 (16%)	63 (44%)
IV	47 (13%)	45 (31%)

## Data Availability

In accordance with the journal’s guidelines, we will provide our data for independent analysis by a selected team by the Editorial Team for the purposes of additional data analysis or for the reproducibility of this study in other centers if such is requested.

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
