# Peer review of "No Racial Disparities Observed Using Point-of-Care Genetic Counseling and Testing for Endometrial and Ovarian Cancer in a Diverse Patient Population: A Retrospective Cohort Study"

_cancers, 2024, doi:10.3390/cancers16081598_

Round 1

Reviewer 1 Report

Comments and Suggestions for Authors

This study investigates genetic counseling and testing rates for patients with gynecologic malignancies at a tertiary care center with a diverse patient population. The findings indicate high rates of genetic counseling and testing, with no apparent racial disparities observed. The authors highlight the increasing utilization of mismatch repair immunohistochemistry (MMR IHC) testing, with a notable rise observed over the study period.

I have the following comments:

1. More details about who are eligible for genetic counseling or genetic testing are needed based on the Society of Gynecologic Oncology guideline. These details can be presented in the Introduction or Methods section.

2. The Methods section is too simplified. The authors need to present more details about the study. For example, explanation about ‘a two-part retrospective cohort study’, how many cases were excluded due to without pathological confirmation of diagnoses.

3. Figure 1 should start with 373 cases of endometrial cancer. A figure may be presented for ovarian cancer.

4. Multivariate logistic models may be used to assess factors associated with receiving MMR, genetic counseling, and genetic testing.

5. The manuscript could benefit from a more detailed discussion of potential factors contributing to the successful implementation of genetic counseling and testing services.

Author Response

Thank you sincerely for taking the time to review the manuscript. 

  1. More details about who are eligible for genetic counseling or genetic testing are needed based on the Society of Gynecologic Oncology guideline. These details can be presented in the Introduction or Methods section.

Additional line was added further explaining family history criteria (45-47). SGO criteria was also added in the Discussion section (193-197).

  1. The authors need to present more details about the study. For example, explanation about ‘a two-part retrospective cohort study’, how many cases were excluded due to without pathological confirmation of diagnoses.

Details are added to Methods (98-105)

  1. Figure 1 should start with 373 cases of endometrial cancer. A figure may be presented for ovarian cancer.

Figure 1 modified and Figure 4 added as suggested, respectively.

  1. Multivariate logistic models may be used to assess factors associated with receiving MMR, genetic counseling, and genetic testing.

Using Fisher’s exact test on the data stratified by genetic counseling, by genetic testing, by MMR done etc. for endometrial and ovarian cancers, very few of the variables were found to have a statistically significant difference between the two comparison groups. For example, for ovarian cancer the only significant variable was staging (i.e. no association with race, ethnicity, language, insurance type, BMI, or family history of cancer). It is likely there would be little association in a logistic model in this setting.

  1. The manuscript could benefit from a more detailed discussion of potential factors contributing to the successful implementation of genetic counseling and testing services.

Paragraph on point of care services (“mainstreaming”) added (lines 272-280), the most likely contributor to successful rates.

Reviewer 2 Report

Comments and Suggestions for Authors

The authors present a study examining the racial disparities in genetic testing and counselling in an under represented population.  The study is well written, and results seem sound.  A few minor points to strengthen the manuscript.

1. The authors should update their terminology on race/ethnicity on recent dialogues on this topic.  For example PMID 38470200; 38050060

2. The authors should italicize the genes, and also use variants instead of mutation (PMID 27657676)

Author Response

Thank you sincerely for taking the time to review the manuscript. 

  1. The authors should update their terminology on race/ethnicity on recent dialogues on this topic.  For example PMID 38470200; 38050060

These references were a very interesting read. The way our epic EMR collects ethnicity is unfortunately dated, and only collects information on whether they are “Hispanic” or “Not Hispanic or Latino or Spanish Origin”. It is also unfortunately a similar limited classification system for race. Further specified in Methods (line 110-113).

  1. The authors should italicize the genes, and also use variants instead of mutation (PMID 27657676)

Language changed as recommended.

Reviewer 3 Report

Comments and Suggestions for Authors

The authors describe a retrospective study examening wether patients with gynaecologic cancer receive genetic counseling regardsless of racial background.

The manuscript is well written and I think the subject of this study is relevant.

However, I have a problem with  the fact that almost all  counseling was performed by a gynaecologic oncologist ( line 34: 93%; line 143: 98%; line 163: 93%) and that the authors make no comment whatsoever on this. What is the quality of the genetic counseling? Are the oncologist trained for this? Why were the patients not referred to a clinical geneticist?

In a published article about genetic counseling in patients with gynaecologic cancer in Europe access to a genetic counselor was reported by 87% of providers and 55% deferred all testing to genetic counselors  ref; J  Genet  Couns, 2018 Feb:27(1):177-186. 

Clinical geneticists are trained to perform genetic counseling and testing, oncologists are not.

The authors have to comment on this in the manuscript.

Author Response

Thank you sincerely for taking the time to review the manuscript. 

  1. However, I have a problem with the fact that almost all  counseling was performed by a gynaecologic oncologist ( line 34: 93%; line 143: 98%; line 163: 93%) and that the authors make no comment whatsoever on this. What is the quality of the genetic counseling? Are the oncologist trained for this? Why were the patients not referred to a clinical geneticist? In a published article about genetic counseling in patients with gynaecologic cancer in Europe access to a genetic counselor was reported by 87% of providers and 55% deferred all testing to genetic counselors  ref; J  Genet  Couns, 2018 Feb:27(1):177-186. Clinical geneticists are trained to perform genetic counseling and testing, oncologists are not.

Thank you for this point and the opportunity to clarify. Patients that tested positive for a genetic variant were additionally provided genetic counseling by a trained genetic counselor as well (this number was not included in the 93% value mentioned above). Preliminary counseling was mostly performed by a gynecologic oncologist, where testing was offered at the same time. This has been updated in methods line 114-117.

Round 2

Reviewer 1 Report

Comments and Suggestions for Authors

The authors have fully addressed the reviewers' comments. The manuscript is now sufficiently improved to warrant publication in Cancers.

Reviewer 3 Report

Comments and Suggestions for Authors

no other comments